# Updated Prediction of Air Quality Based on Kalman-Attention-LSTM Network

**Hao Zhou** [1,*,†], **Tao Wang** [2,†], **Hongchao Zhao** [1,†] **and Zicheng Wang** [3]

1. College of Optoelectronic Engineering, Chongqing University, Chongqing 400044, China
2. School of Astronautics, Harbin Institute of Technology, Harbin 150080, China
3. College of Intelligent Systems Science and Engineering, Harbin Engineering University, Harbin 150001, China
* Correspondence: 202108021064t@cqu.edu.cn
† These authors contributed equally to this work.

**Abstract:** The WRF-CMAQ (Weather research and forecast-community multiscale air quality) simulation system is commonly used as the first prediction model of air pollutant concentration, but its prediction accuracy is not ideal. Considering the complexity of air quality prediction and the high-performance advantages of deep learning methods, this paper proposes a second prediction method of air pollutant concentration based on the Kalman-attention-LSTM (Kalman filter, attention and long short-term memory) model. Firstly, an exploratory analysis is made between the actual environmental measurement data from the monitoring site and the first forecast data from the WRF-CMAQ model. An air quality index (AQI) was used as a measure of air pollution degree. Then, the Kalman filter (KF) is used to fuse the actual environmental measurement data from the monitoring site and the first forecast results from the WRF-CMAQ model. Finally, the long short-term memory (LSTM) model with the attention mechanism is used as a single factor prediction model for an AQI prediction. In the prediction of $O_3$ which is the main pollutant affecting the AQI, the results show that the second prediction based on the Kalman-attention-LSTM model features a better fitting effect, compared with the six models. In the first prediction (from the WRF-CMAQ model), for the RNN, GRU, LSTM, attention-LSTM and Kalman-LSTM, SE improved by 83.26%, 51.64%, 43.58%, 45%, 26% and 29%, respectively, RMSE improved by 83.16%, 51.52%, 43.21%, 44.59%, 26.07% and 28.32%, respectively, MAE improved by 80.49%, 56.96%, 46.75%, 49.97%, 26.04% and 27.36%, respectively, and R-Square improved by 85.3%, 16.4%, 10.3%, 11.5%, 2.7% and 3.3%, respectively. However, the prediction results for the Kalman-attention-LSTM model proposed in this paper for other five different pollutants ($SO_2$, $NO_2$, $PM_{10}$, $PM_{2.5}$ and CO) all have smaller SE, RMSE and MAE, and better R-square. The accuracy improvement is significant and has good application prospects.

**Keywords:** second prediction; AQI; Kalman filter; Kalman-attention-LSTM

## 1. Introduction

### 1.1. Background Information

The practice of pollution prevention and control shows that it is one of the most effective methods in reducing the harm inflicted by air pollution on human health and the environment and for improving the ambient air quality to establish the air quality forecast model, which allows us to know the possible air pollution process in advance and to take corresponding control measures. At present, air quality assessment methods based on simulated meteorological field information and a pollutant emission inventory include the Community Multiscale Air Quality Model (CMAQ), the Operational Street Pollution Model (OSPM), the Nested Air Quality Prediction Modeling System (NAQPMS), etc. Among them, the Weather Research and Forecasting-Community Multi-scale Air Quality Simulation System (WRF-CMAQ model) is a common method used to predict air quality.

However, the existing air quality prediction methods cannot effectively predict the data with complex affecting factors such as the AQI. Due to the influence of complex mechanism and spatial diffusion, the accuracy of the first-round prediction based on the WRF-CMAQ model is not ideal, leading to a large prediction error. In order to improve the prediction accuracy, we introduced the concept of the second prediction. The essence of the second prediction is to improve the prediction accuracy and obtain smaller RMSE and MAE based on the results of the first prediction and other time series information through reasonable algorithms. Moreover, the effectiveness of the WRF-CMAQ model is limited by traditional deterministic methods based on the use of default parameters and a lack of actual observations. Therefore, prediction methods based on time series data came into being, such as traditional machine learning methods and time series prediction models.

The traditional regression model cannot have an ideal performance in the prediction of the influence of uncertain factors; however, with the introduction of a hidden layer in neural network model, this makes it to the mainstream choice for solving the problem of complex linear prediction, whereas the LSTM network is the mainstream choice for solving the problem of long sequence, as well as having the effect of the breakthrough. At present, various LSTM-based models such as the attention-LSTM, the BiLSTM and the CNN-LSTM are widely used in the prediction of the long time series. Generally, the pollutant prediction system is a linear, discrete and time-varying system with accurate and calculable information at each time step. Both the measured value and the first predicted value of each time step contain white noise. The above two points indicate that the data characteristics of the system conform to the application standard of the Kalman filter. At the same time, the most appropriate method in this system is the Kalman filter, because it can solve the linear filtering problem with a recursive method in order to predict the system state from the observation signals and external inputs containing noise. Inspired by the idea of solving the optimal state estimation of the system in cybernetics, we introduce the classical Kalman filter into the attention-LSTM model, aiming to improve the accuracy and reliability in dynamic forecast using data correction, which makes the prediction accuracy and long-term stability of the Kalman-attention-LSTM model in this paper better than the traditional LSTM model and attention-LSTM model.

What we are presented with are the first-round prediction results (from WRT-CMAQ system) and the measured data (from monitoring site). Based on the above two sets of data, the second-round prediction of pollutant concentration is carried out by using the Kalman-LSTM-attention model proposed in this paper. The second-round prediction makes up for the low accuracy of the first-round prediction.

### 1.2. Related Works

LSTM was proposed by Hochreiter and Schmidhuber [1] in 1997 to alleviate the vanishing gradient problem of the RNN to a certain extent. Recently, as a result of the rapid increase in the number of measured data, artificial intelligence techniques have been intensively used in predicting air quality as an alternative to the traditional models in the field of air quality prediction. Additionally, researchers began to shift their research focus to hybrid models, hoping to obtain a higher prediction accuracy than with traditional models [2,3]. The deep learning method has achieved ideal results in the regional meteorological data set, which has also been verified in this paper. Akbal et al. [4] proved that the hybrid model which consists of the FNN, CNN and LSTM has the best predictive accuracy for particulate matter (PM). Most of the time series prediction papers based on the RNN model have the mixed LSTM model [5,6] or introduced a gate mechanism similar to the LSTM model [7,8], which proves that the LSTM model is successful in relation to the time series prediction problem. In meteorological applications, Krishan et al. [9] predicted O, $PM_{2.5}$, NO and CO concentrations at a site in Delhi based on the LSTM method; Tsokov et al. [10] proposed a deep spatiotemporal model based on the 2D CNN and LSTM, which used a genetic algorithm to automatically select input variables and optimize hyperparameters for air pollution prediction. Qadeer et al. [11] predicted $PM_{2.5}$ concentration in two big cities

in South Korea based on the Bi-directional LSTM (BiLSTM), and the results were better than other traditional gradient tree enhancement models with cyclic and convolutional neural networks. Jiao et al. [12] used the LSTM model to predict the AQI through temperature, $PM_{2.5}$, $PM_{10}$, $SO_2$, wind direction, $NO_2$, CO and $O_3$, proving better than the linear regression prediction method.

The LSTM alleviates the gradient vanishing problem of the RNN to a certain extent, while the attention mechanism becomes an effective means for solving the vanishing gradient and gradient explosion problems of the RNN. In the past decade, the attention mechanism has been applied to the optimization of neural networks [13]. Shi et al. [14] proposed a long short-term memory network model based on spatial attention (SA-LSTM), which combined LSTM and a spatial attention mechanism to adaptively use multi-factor spatio-temporal information in order to predict the concentration of air pollutants. Yuan et al. [15] designed a multi-attention mechanism based on multi-layer perception, including monitoring point attention, temporal feature attention and weather attention, in order to obtain the spatio-temporal and meteorological dependence of $PM_{2.5}$, and proposed a hybrid deep-learning method based on a multiple attention LSTM (MAT-LSTM) neural network for $PM_{2.5}$ concentration prediction. Liu et al. [16] proposed a wind sensitive attention mechanism based on the LSTM model in order to predict air pollution by considering the influence of wind direction and wind speed on spatial and temporal variations of $PM_{2.5}$ concentration in neighboring areas. The proposed method outperforms the multilayer perceptron, support vector regression, LSTM neural network and extreme gradient boost algorithm in predicting $PM_{2.5}$ concentration. Chen et al. [17] proposed a double LSTM prediction model based on the attention mechanism. EXtreme Gradient Boosting (XGBoost) regression was used to construct the optimal promotion tree, and the optimal prediction results were obtained by combining the single factor model and the multi-factor model.

A Gated Recurrent Unit (GRU) is a common variant of the LSTM. By simplifying the gate mechanism of the LSTM, it makes the training more convenient with fewer parameters. Sonawani et al. [18] proposed a GRU model to estimate and monitor the $NO_2$ pollutants in Pune, India, by evaluating and optimizing the model based on the number of features, number of neurons, number of retrospections and number of eras. Air pollution forecasts can provide reliable information on future air pollution conditions, which can facilitate the effective operation of air pollution control and the development of prevention plans. Tao et al. [19] proposed a Convolutional Bidirectional Gated Recurrent Unit (CBGRU) method based on the combination of a one-dimensional convolutional neural network and a bidirectional GRU neural network, and they used the Beijing $PM_{2.5}$ dataset in the UCI machine learning library for example analysis. Zhou et al. [20] took hourly $PM_{2.5}$ concentration information and weather information from Beijing as their input and based on the GRU model, trained four models according to the four seasons, spring, summer, autumn and winter, and verified the feasibility of this method. However, most of the papers based on the GRU deliberately avoid the effect comparison with the LSTM model, and the work of Liu et al. [21] shows that the GRU model is slightly inferior to the LSTM model in terms of long-term accuracy.

As a widely used hybrid model, the CNN-LSTM combines the respective advantages of the CNN and LSTM, with the CNN being able to effectively extract the features of grid data, and the LSTM being able to effectively process time series data [22]. In Stefan et al. [10], a neural network is presented based on a two-dimensional convolution and the long short-term memory network model of time and space, using the genetic algorithm to automatically choose the input variables and allow the optimization of parameters; multiple sites in Beijing air quality data sets for the experimental results show the proposed air pollution prediction model with a good consistency in time and space prediction results. Wang et al. [23] proposed a CNN-BiLSTM-attention model to predict the AQI. This model used the CNN to extract the features and influences of the input data and improved the accuracy of the AQI prediction. Gilik et al. [24] combined the convolutional neural network with the long short-term memory deep neural network model to predict

the concentration of air pollutants in multiple locations within the city by using the spatio-temporal relationship. In terms of transfer learning, as the network was transferred from Kocali to Istanbul, the model showed a more accurate prediction performance. Li et al. [25] developed a hybrid CNN-LSTM model for predicting $PM_{2.5}$ concentration in the next 24 h in Beijing, making full use of the advantages of the CNN in effectively extracting air quality related features and the LSTM in reflecting the long-term historical process of the input time series data.

Inspired by the idea of solving the optimal state estimation of the system in cybernetics, in order to predict the state of the system from noisy observation signals and external inputs, some researchers began to introduce the classical Kalman filter into the timing prediction. Song et al. [26] proposed an air quality assessment method based on the LSTM-Kalman model, which applied the Kalman filter to the LSTM model and was superior to the independent Kalman filter and the independent LSTM. Li et al. [27] proposed a KLS algorithm combining the Kalman filter (KF), LSTM and support vector machine (SVM) and adopted statistical filtering and deep learning algorithms to achieve the fusion of time series prediction and variable regression.

In addition, there are many other hybrid models for the LSTM. Wu et al. [28] proposed a VMD-LSTM model combining the VMD and LSTM to predict the AQI, which has a high prediction accuracy for AQI class, and which is what the BP and LSTM models cannot achieve. Zhou et al. [29] proposed a deep multi-output LSTM (DM-LSTM) neural network model, which combined three deep learning algorithms (minibatch gradient descent, dropout neuron and L2 regularization) to extract key factors of complex spatio-temporal relationships and reduce error accumulation and propagation in multi-step-ahead air quality prediction. The spatial and temporal stability and accuracy of regional multi-step-ahead air quality prediction are both significantly improved. Chang et al. [30] proposed an aggregated LSTM model (ALSTM) on the basis of the LSTM model, which aggregated the three LSTM models (the local air quality monitoring station, the nearby industrial area monitoring station and the external pollution source monitoring station) into a prediction model. Early predictions are based on information from external sources of pollution and nearby industrial air quality monitoring stations. Qi et al. [31] figured graph convolutional networks and the LSTM and put forward a model of the GC-LSTM; the historical observation data of different stations were constructed as a spatio-temporal map sequence, whilst the historical air quality variables, meteorological factors, spatial terms and temporal attributes were defined as map signals to model and predict the spatio-temporal variation of $PM_{2.5}$ concentration. Zhao et al. [32] proposed a LSTM fully connected (LSTM-FC) neural network model. In this model, temporal simulators based on the LSTM model were used to simulate local changes in $PM_{2.5}$ pollution, and spatial combinations based on neural networks were used to capture the spatial correlation between $PM_{2.5}$ pollution in central stations and neighboring stations, with the model outperforming the ANN and LSTM models on the same dataset. At the same time, Cheng et al. [33] proposed a novel data assimilation (DA) technique intending to incorporate real-time observations from different physical spaces, which is the one of the current observational methods used to perform variational DA with a low computational cost. Also, Zhuang and Cheng et al. [34,35] demonstrated that system efficiency can be improved through the combination of reduced-order modeling and recurrent neural network models. Data assimilation enables the system to adjust the simulation results according to the observed data.

In the previous literature, we noticed that there is no pollutant concentration prediction model for the second prediction at present. Although many optimization methods based on the LSTM model have emerged to improve the prediction accuracy, it is still a rare choice to introduce the Kalman filter and the attention mechanism into the LSTM model. In order to fill the research gap and further improve the model accuracy, this paper established a Kalman-attention-LSTM model for predicting air pollution concentration by combining the Kalman Filter, attention mechanism, and LSTM.

*1.3. Chapters Arrangement*

The remaining papers are organized as follows: Section 2 introduces the model building and optimization method; Section 3 discusses and analyzes the prediction results of the model; and Section 4 is the research conclusion.

## 2. Materials and Methods

*2.1. Data Collection and Preprocessing*

In order to conduct second prediction based on the first prediction, we acquired the downtown monitoring of air quality forecast basic data for a long period of time in Chongqing Municipality, China. It includes forecast pollutant concentration data, meteorological data, forecast-measured meteorological data and the measured data of pollutant concentration. The time span for all first forecast data is from 23 July 2020 to 13 July 2021, and the time span for all measured data is from 16 April 2019 to 13 July 2021. The daily forecast time is fixed at 7 a.m., with the measured data of the day as well as the first forecast data of the day able to be obtained at 7 a.m. or before (the forecast time range goes up to 11 p.m. on the third day). Due to the limitations concerning the authority of the monitoring data and the functions of corresponding monitoring equipment, the measured data of some meteorological indicators cannot be obtained. Due to the high accuracy of the first forecast in relation to the adjacent date, the accuracy of the second forecast in relation to the adjacent date is also high.

However, after browsing the daily and hourly measured data, we found the following in the hourly pollutant concentration of the monitoring site and the measured meteorological data: the overall data of some hours (possibly continuous or discontinuous) were lost from 0:00 to 23:00 in one day; the data pre-processing included data integrity discrimination and deletion, data vacancy filling, and data normalization; after pre-processing, it is necessary to check whether daily and hourly data dates can be corresponding and to align the days.

Due to the unknown working condition of the monitoring site in Chongqing Municipality, it is difficult to restore the real pollutant concentration using the average value method of adjacent points. In this case, data vacancies of the hours mentioned above should be replaced by data calculated using the Lagrange interpolation method. When the integrity of the data is over 80%, the data will be retained and the Lagrange interpolation method will be adopted. If the data integrity requirements are not met, delete all rows to improve data group reliability.

In Missing Completely at Random (MCAR) hypothesis, the cause of missing data is independent of observed and unobserved variables. In the Missing at Random (MAR) hypothesis, the reason for missing data depends on fully observed covariates and has nothing to do with unobserved factors. For the loss of concentration data of one or more pollutants in a certain hour, Lagrange's interpolation method is adopted to construct a set of first functions, represented as

$$
\begin{aligned}
l_i(x) &= \frac{(x-x_0)\cdots(x-x_{i-1})(x-x_{i+1})\cdots(x-x_n)}{(x_i-x_0)\cdots(x_i-x_{i-1})(x_i-x_{i+1})\cdots(x_i-x_n)} \\
&= \prod_{\substack{j=0 \\ j \neq 1}}^{n} \frac{x-x_j}{x_i-x_j}, \ (i=0,1\cdots,n)
\end{aligned}
\tag{1}
$$

$$
l_1(x_j) = f(x) = \begin{cases} 0, & j \neq i \\ 1, & j = i \end{cases}
\tag{2}
$$

$$
L_n(x) = \sum_{i=0}^{n} y_i l_i(x) = \sum_{i=0}^{n} y_i \left( \prod_{\substack{j=0 \\ j \neq i}}^{n} \frac{x-x_i}{x_i-x_j} \right)
\tag{3}
$$

where $l_i(x)$ is the n-degree polynomial, $x_i$ is the number of days, and $L_n(x)$ is the concentration of pollutants on a certain day.

Since the data obtained from the monitoring site contain two kinds of time granularity, one is daily data and the other is hourly data. When the proportion of missing data with time granularity is large, the reliability of the corresponding daily data provided by the monitoring site becomes doubtful. Therefore, we do not use controversial daily data, but choose to use Lagrange interpolation method to fill up the hourly data and generate more reasonable daily data. The Lagrange interpolation code was compiled using PyCharm and interpreted using Python 3.8.2. Figure 1 shows the testing effect of interpolating selected data. If the curve fitting is carried out on the data, the curve is smooth, and the interpolation effect meets the requirements of data pretreatment and data filling.

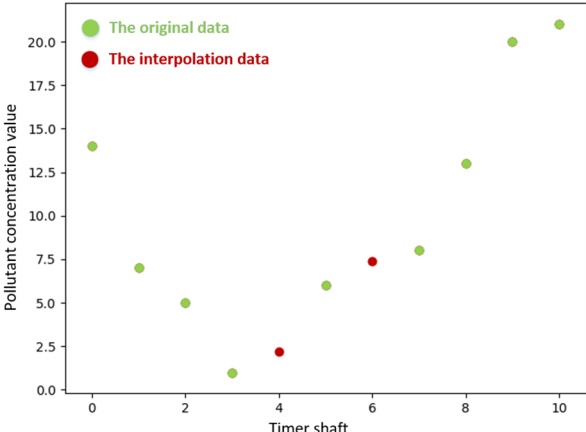

**Figure 1.** Lagrange's interpolation in small quantities.

Data analysis found different concentration of pollutants at the same unit of measurement, but that there are many differences between orders of magnitude. Meteorological data exist as indicators of difference in the unit of measurement and scales, so it is necessary to standardize the data processing, using the method of maximum-minimum value for some orders of magnitude difference during bigger, normalized processing. The purpose is to eliminate the difference between the orders of magnitude of data in each dimension. According to (4), $x_{min}$ represents the minimum value, $x_{max}$ represents the maximum value and $x_k$ represents the normalization result. It represents as

$$x_k = \frac{(x_k - x_{min})}{x_k - x_{max}} \tag{4}$$

*2.2. Modeling and Optimization*

2.2.1. First Forecast Source

WRF-CMAQ (Weather research and forecast-community multiscale air quality) simulation system is commonly used as the first prediction model of air pollutant concentration. WRF-CMAQ model mainly consists of WRF and CMAQ. WRF is a mesoscale numerical weather prediction system, which is used to provide weather field data for CMAQ; WRF structure of the mesoscale numerical weather prediction system is shown in Figure 2. CMAQ is a three-dimensional Euler atmospheric chemistry and transport simulation system; CMAQ structure of air quality prediction and assessment system are shown in Figure 3. Based on the meteorological information from WRF and the pollution emission inventory in the field, it simulates the change process of pollutants based on the principle of physical and chemical reactions and then obtains the forecast results at specific time points or time periods.

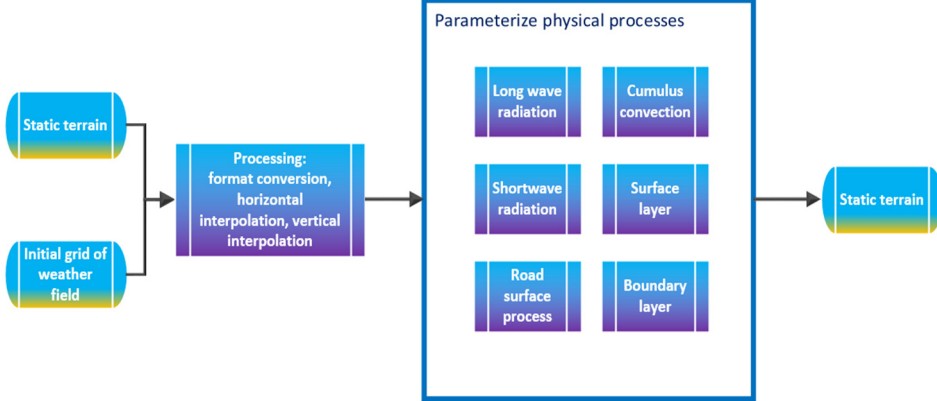

**Figure 2.** WRF structure of mesoscale numerical weather prediction system.

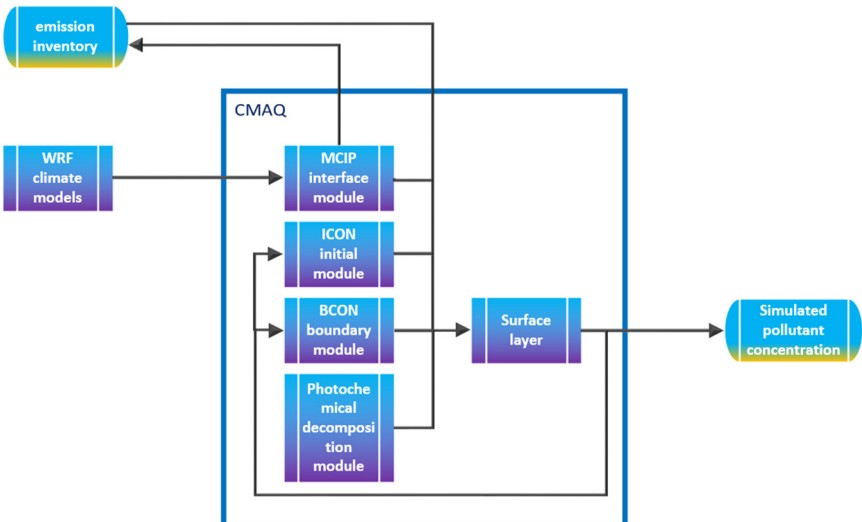

**Figure 3.** CMAQ structure of air quality prediction and assessment system.

This WRF-CMAQ model will be used as the first prediction model of this paper, together with the measured data of monitoring sites, as the data source and research basis. The meteorological information of Chongqing Municipality, China, which are obtained by us, has already included the first prediction results of WRF-CMAQ system. We only need to conduct basic normalization and completion operations on its data.

2.2.2. Correlation Coefficient and Clustering of Variables

Air Quality Index (AQI) is usually used to measure daily pollution. $AQI$ is calculated based on $IAQI$; $IAQI_P$ is the air quality index of a kind of pollutant $P$. $C_P$ is the mass concentration value of pollutant $P$. $BP_{Hi}$ and $BP_{Lo}$ are high and low values of contaminant concentration limit similar to $C_P$. $IAQI_{Hi}$ and $IAQI_{Lo}$ are the air quality sub-index corresponding to $BP_{Hi}$ and $BP_{Lo}$. The maximum value of the $AQI$ is calculated as

$$IAQI_P = \frac{IAQI_{Hi} - IAQI_{Lo}}{BP_{Hi} - BP_{Lo}} \cdot (C_P - BP_{Lo}) + IAQI_{Lo} \tag{5}$$

$$AQI = max\{IAQI_1, IAQI_2, IAQI_3, \dots, IAQI_n\} \tag{6}$$

$$AQI_{max} = max\left\{ \begin{array}{c} IAQI_{SO_2}, IAQI_{NO_2}, IAQI_{PM_{10}}, \\ IAQI_{PM_{2.5}}, IAQI_{O_3}, IAQI_{CO} \end{array} \right\} \tag{7}$$

The correlation coefficient is a statistical indicator reflecting the closeness of correlation between variables. Yet we have five weather variables and six pollutant variables. In addition to the influence of one meteorological condition on one pollutant, there is also the

influence of one pollutant on another, and the influence of meteorological conditions on another. Therefore, it is necessary to independently calculate the influence between the two variables and predict that the concentration of some pollutants is the main influencing factor of AQI. In order to achieve this, assume that the observation matrix of the sample represents as

$$X = \begin{bmatrix} x_{11} & x_{12} & \cdots & x_{1p} \\ x_{21} & x_{22} & \cdots & x_{2p} \\ \cdots & & \cdots & \cdots \\ x_{n1} & x_{n2} & \cdots & x_{np} \end{bmatrix} \tag{8}$$

$$x_{ij}^* = \frac{x_{ij} - \overline{x}_j}{\sqrt{Var(x_j)}} \ (i = 1,2,3 \cdots n \ j = 1,2,3 \cdots p) \tag{9}$$

$$\overline{x}_j = \frac{1}{n} \sum_{i-1}^{n} x_{ij}, Var(x_j) = \frac{1}{n-1} \sum_{i-1}^{n} (x_{ij} - \overline{x}_j)^2 \ (j = 1,2,3 \cdots p) \tag{10}$$

Thus, the correlation coefficient matrix represents as

$$R = \begin{bmatrix} r_{11} & r_{12} & \cdots & r_{1p} \\ r_{21} & r_{22} & \cdots & r_{2p} \\ \cdots & & \cdots & \cdots \\ r_{n1} & r_{n2} & \cdots & r_{np} \end{bmatrix} \tag{11}$$

$$r_{ij} = \frac{Cov(x_i, x_j)}{\sqrt{Var(x_1)}\sqrt{Var(x_2)}} = \frac{\sum_{k=1}^{n}(x_{ki} - \overline{x})(x_{ki} - \overline{x}_j)}{\sqrt{\sum_{k=1}^{n}(x_{ki} - \overline{x}_i)^2}\sqrt{\sum_{k=1}^{n}(x_{ki} - \overline{x}_j)}} \tag{12}$$

The influence of meteorological conditions on pollutant diffusion or settlement should be analyzed according to the influence of various meteorological features on the rise or decline of the AQI. K-means clustering algorithm is one of the most common clustering methods; it calculates the best category based on the similarity of the distance between points, with the data divided into the same cluster having similarity. All meteorological data and pollutant concentration data were normalized from between 0 to 1 before clustering. K-means clustering algorithm needs to randomly select two centroids from the sample of the same pollutant concentration and meteorological conditions as the initial cluster center; one center of mass represents a class. $\mu_1^{(0)}$ and $\mu_2^{(0)}$ are the center of mass, $J(c, u)$ represent the clustering effect, $x_i$ is the sample point position. This is represented as

$$J(c, u) = min \sum_{i=1}^{M} ||x_i - \mu_{c_i}||^2 \tag{13}$$

The classification is based on the distance from the sample point to the center of mass of the cluster in which it is located; whoever is closer is in the same category as the data center. The most common method is to calculate the Euclidean distance from each remaining sample point to each center of mass, which is ordinary two-dimensional data, based on the Pythagorean theorem and represented as

$$d(x, \mu) = \sqrt{\sum_{i=1}^{n}(x_i - \mu_i)^2} \tag{14}$$

where $d(x, \mu)$ is the Euclidean distance. Start the loop and group them into the cluster with the center of mass least distant from each other. This is represented as

$$c_i^t < -\text{arg}min||x_i - \mu_k^t||^2 \tag{15}$$

where $c_i^t$ represents the classification of sample points. After all the sample data points were divided into clusters, the centroid of each new cluster was calculated using the average distance between the sample points and the cluster and represented as

$$\mu_k^{(t+1)} < -\arg min \sum_{i:c_i^t=k}^{b} ||x_i - \mu||^2 \tag{16}$$

where $\mu_k^{(t+1)}$ represents the average distance between the sample point and the cluster. Repeat the above steps to calculate the Euclidean distance iteratively and re-divide all sample points. The sum of the squares of the distances from all sample points of a cluster to the center of mass is $\sum_{j=0}^{m} \sum_{i=1}^{n} (x_i - \mu_i)^2$. When the number of iterations reaches the maximum or the centroid does not change anymore, it means convergence has been achieved and the clustering has ended.

### 2.2.3. Kalman Filter

Now there are two sets of data, one is the predicted pollutant concentration data, and the other is measured pollutant concentration data. Considering the dynamic characteristics of the tested system, the data sources are reasonable, but there are also noises and errors in the acquisition process of variables, so the form of prediction + correction is used to make the optimal estimation. Kalman filter is essentially an optimized autoregressive data processing algorithm that does not require all previous data. Kalman filter can predict the next step of a dynamic system with uncertain information under the interference of noise information. To put it simply, Kalman filter mainly includes two steps: state variable estimation and state variable correction. The specific mathematical modeling process is as follows.

Firstly, the predicted value of the pollutant concentration (this predicted concentration is not an optimal prediction) at the current time is estimated from the predicted value of the pollutant concentration in the previous hour combined with the external control. The state prediction equation presents as

$$x_t^- = Ax_{t-1} + Bu_{t-1} \tag{17}$$

where $x_{t-1}$ represents the predicted value of the previous hour and uses the first prediction result from WRF-CMAQ model as input, $u_{t-1}$ represents the external control input, $x_t^-$ represents the predicted value of the current moment and is also called a prior state estimate. In Kalman filter design, both state transition matrix $A$ and control matrix $B$ are determined by the properties of the system. $A$ represents the state transfer matrix from the previous hour to the current moment. Due to the time series information being one-dimensional, we use the scalar Kalman filter which means the actual form of state transfer matrix $A$ is the scalar, $A$ equal to 1. $B$ represents the control matrix, with the control matrix being used to convert external control inputs into state information. However, in the actual situation, the update of pollutant concentration status is not controlled by humans, which means gain of control is not necessary, and therefore $B$ chose 0.

Then, the covariance matrix of the previous hour is used to predict the current covariance matrix, represented as

$$P_t^- = AP_{t-1}A^T + Q \tag{18}$$

where $Q$ represents the mean square error matrix of process noise and reflects the error between the state transition matrix and the actual process, $P_{t-1}$ is the posterior estimation of covariance an hour before, $A^T$ is the transpose matrix $A$, and $P_t^-$ is the priori estimated covariance at the current time and also the intermediate calculation result of the filter.

The difference between the current measure concentration value and the predicted concentration value is used to correct the predicted value of the current time. State update equation represents as

$$x_t = x_t^- + K_t(z_t - Hx_t^-) \tag{19}$$

where $z_t$ is the real measure value and is also used as the input of real measure results from monitoring sites, with it being one-dimensional time-serious information with the time granularity as 1 h. $H$ is observer matrix and it is used to convert the measured value to correspond to the state variable; $H$ chose 1 due to the one-dimensional time series information. Furthermore, $\left(z_t - Hx_t^-\right)$ is the residual of actual measurements and predicted observations, and together with Kalman gain can correct prior predictions. $K_t$ is Kalman gain. $x_t$ is the current optimal state estimate as well as the output value of the Kalman Filter, which is also known as a posteriori state estimate.

Update the Kalman gain with the optimal state estimate at the current time, and the expression of Kalman gain under the minimum mean square error criterion is obtained. This is represented as

$$K_t = P_t^- H^T \left(HP_t^- H^T + R\right)^{-1} \tag{20}$$

All the variables in this formula have been described previously. The Kalman gain determines whether we trust the prediction result more or measure the results more. If we trust the prediction result more, this residual of $\left(z_t - Hx_t^-\right)$ will have less weight.

Finally, find the relationship between $P_t$ and $K_t$ and then get the noise covariance matrix at the current time; this step is designed to prepare for next iteration, and uncertainty of the predicted state is reduced by updating the noise distribution of the best estimator. It updates the forecast error represents as

$$P_t = (I - K_t H)P_t^- \tag{21}$$

where $P_t$ is the current posterior estimation of covariance and $I$ is the identity matrix. The following time, the new noise covariance matrix $P_t$ is used to make a new prediction, and the autoregressive operation of the algorithm is realized.

The purpose of Kalman filter is to solve the optimal state estimation between the one prediction result and the actual measured value, and its principle is to minimize the covariance of the optimal state estimation and make it get closer and closer to the real value. The core of the Kalman filter is the computing of Kalman gain, which reflects the model prediction error during the optimal state estimation process. The Kalman filter gives the data different proportions according to the accuracy of the data, and the data with higher accuracy has a higher proportion. The optimal state estimation of Kalman filter is then output by computing the first prediction values and the actual monitoring data according to the Kalman gain fusion. The optimal state estimation value of Kalman filter then corrects the covariance of the previous prediction process and calculates iteratively to obtain the filtering results.

The input data of Kalman filter contains two groups of time series data. The first group is the first prediction result generated based on WRF-CMAQ model provided by the monitoring site and input into $x_{t-1}$; the other group is the real measure data from the monitoring site and input into $z_t$. The effect of Kalman filter is to correct the first prediction pollutant concentration data with real-measure pollutant concentration data. The structure of the LSTM network with attention mechanism will be introduced later.

### 2.2.4. LSTM Network

Long Short-Term Memory (LSTM) network is a chain-structure-improved network based on the RNN model, with the core characteristics of time memory and cyclic adjustment of training feedback. The biggest difference between LSTM and RNN networks is four neural network layers are used in one cell in LSTM. Furthermore, internal interaction modes are added and three gate structures are added, including forget gate, input gate, and output gate. The cell structure of the LSTM is shown in Figure 4, and the LSTM network model is shown in Figure 5.

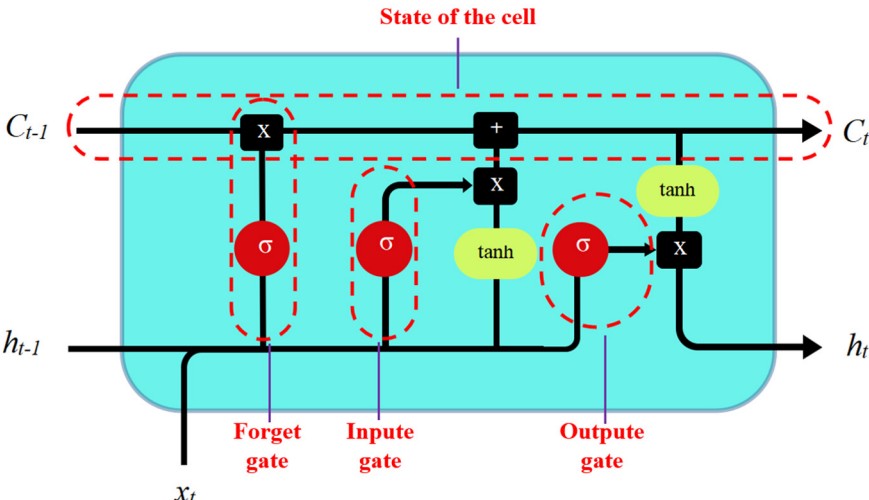

**Figure 4.** LSTM cell structure.

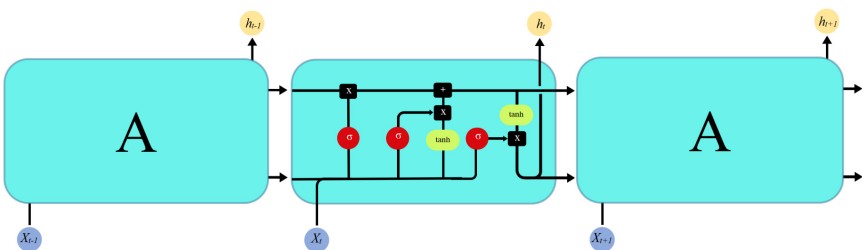

**Figure 5.** LSTM network structure.

The forget gate contains a sigmoid network layer and a bitwise multiplication operation. The sigmoid layer is responsible for screening the combined input signals of $x_t$ at the current moment and $h_{t-1}$ at the last moment. $f_t$ represents the forget gate at the time step $t$. The function of this gate is to output a signal from 0 to 1 through sigmoid multiplied using the state $C_{t-1}$ at the previous time to describe how much the input signal is through. The subscripts of $W$ and $b$ indicate, respectively, the weight and the bias for three different gates. For example, $W_f$ is the weight of input $x_t$ at the gate $f_t$. This formula is represented as

$$f_t = sigmoid\left(W_f[h_{t-1}, x_t] + b_f\right) \tag{22}$$

The second gate is the input gate $i_t$ at the time step $t$. The input gate is responsible for screening the reserved part of the combined input signal of $x_t$ at the current moment and $h_{t-1}$ at the last moment. It contains a sigmoid layer and a tan h network layer. The sigmoid layer effect is the same as that in the forget gate. Tan h is the hyperbolic tangent function. In the tanh network layer, the current input $x_t$ and the previous output $h_{t-1}$ are directly combined at the end to create a new state vector called $\widetilde{C}_t$, which ranges from $-1$ to 1. The output of sigmoid and tahn are multiplied to determine whether new information is added to the cell state and represents as

$$i_t = sigmoid(W_i[h_{t-1}, x_t] + b_i) \tag{23}$$

$$\widetilde{C}_t = \tan h(W_c[h_{t-1}, x_t] + b_c) \tag{24}$$

where $C_t$ corresponds to the cell unit at the time step $t$. The output of the forget gate is multiplied by the state of the last moment to select forgetting and retaining some information, and then added together with the input gate to obtain the new cell state information, and the updated cell state will continue to be transmitted to the next moment as the state input and represents as

$$C_t = f_t * C_{t-1} + i_t * \widetilde{C}_t \tag{25}$$

The output gate is responsible for transmitting the output signal to the next neuron. $O_t$ represents the output gate at the time step $t$. $h_t$ represents the hidden state at the time step $t$. The combined input signal of $x_t$ at the current moment and $h_{t-1}$ at the previous moment passes through the sigmoid network layer and is multiplied by $C_t$ to obtain $h_t$ with the input signal at the next moment represented as

$$O_t = sigmoid(W_o[h_{t-1}, x_t] + b_o) \tag{26}$$

$$h_t = O_t * \tanh(C_t) \tag{27}$$

### 2.2.5. Attention Mechanism

Since the multi-dimensional auxiliary variables have different effects on the output, they may affect the prediction results. In this paper, the attention mechanism is used to assign weights to the input of different time steps to improve the prediction effect of pollutant concentration. The main principle is to save the intermediate results generated by the LSTM network for the time series input and associate the results with the output values so that the model learns how to selectively focus on the data and assign more reasonable weights to the data. The network structure is shown in Figure 6.

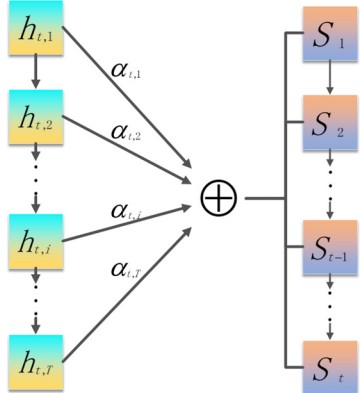

**Figure 6.** The working steps of the attention mechanism.

One way to think about the attention mechanism is to think of the elements in the source as a series of elements about key and values. In this case, an element query in a given target is constructed. By calculating the similarity or correlation between query and each key, the weight coefficient of each key corresponding to value is obtained, and then the weighted sum of values is performed to obtain the final attention value. So essentially, the attention mechanism is a weighted sum of the values of elements in source, while query and key are used to calculate the weight coefficients of the corresponding values. As for the specific calculation process of attention mechanism, if most current methods are abstracted, it can be summarized into two processes: the first process is to calculate the weight coefficient according to query and key, and the second process is to weight and sum the value according to the weight coefficient. The first process can be subdivided into two phases. The first phase calculates the similarity or correlation between query and key; the most common method is to take the dot product of the two vectors and can be represented as

$$e_{t,i} = S_{t-1}^T * h_{t,i} \tag{28}$$

where $S_{t-1}^T$ is the query, $h_{t,i}$ is the key and $e_{t,i}$ represents the similarity between query and key.

In the second phase, the original scores of the first stage are normalized. The score of the first phase is numerically converted using a calculation method similar to SoftMax, and the original calculated score is sorted into the probability distribution with the sum

of the weights of all elements equal to 1. The weight of important elements is highlighted through the internal mechanism of SoftMax and represented as

$$\alpha_{t,i} = Softmax(e_{t,i}) = \frac{\exp(e_{t,i})}{\sum_{k=1}^{T} \exp(e_{t,i})} \tag{29}$$

The LSTM hidden state obtained at time $t$ is $[h_{t,1}, h_{t,2}, \cdots h_{t,i}, \cdots h_{t,T}]^T$; the dot product form is used to calculate the attention weight $\alpha_{t,i}$ of the hidden layer state $h_i$ in accordance with the output at time $t$. Then add the weights to get the attention value $S_t$, represented as

$$S_t = \sum_{i=1}^{T} \alpha_{t,i} h_{t,i} \tag{30}$$

The attention mechanism is added into LSTM neural network, aiming to calculate the weight of each hidden layer state of the network, and the measurement model of Kalman-attention-LTSM is established to predict pollutant concentration.

### 2.2.6. Kalman-Attention-LSTM Network

The complete prediction model is Kalman-attention-LSTM which contains two groups of time series data input. The first group is the first prediction result generated based on WRF-CMAQ model provided by the monitoring site, and the other group is the actual monitoring data of the monitoring site. The input data of the Kalman-attention-LSTM are the two temporal data groups mentioned above. The two temporal data groups mentioned above are also taken as the training and prediction sample of Kalman-attention-LSTM. The network layer structure based on the Kalman-attention-LSTM is shown in Figure 7, which mainly consists of four parts.

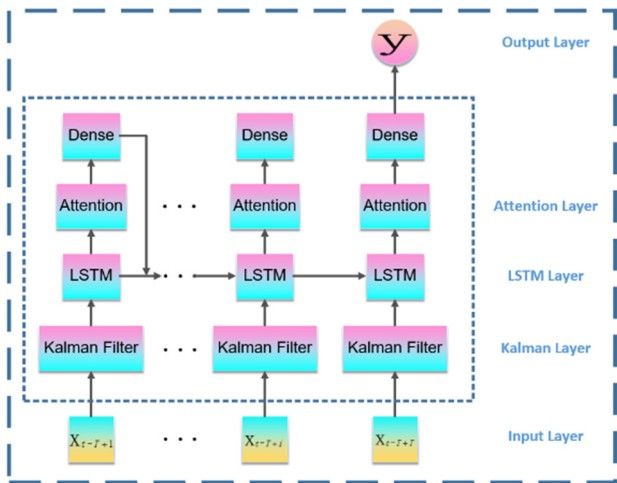

**Figure 7.** Architectures of the Kalman-attention-LSTM network.

Kalman layer: input is auxiliary variable time series and pollutant concentration data processed using Kalman Filter. Set $T$ as the number of time-step smoothing windows, then the input sequence at time $t$ is $[x_{t-T+1}, x_{t-T+2}, \cdots x_{t-T+i} \cdots x_t]^T$.

The most important hyperparameter selections of Kalman filter are matrix Q and matrix R, which are usually given by manual experiments. In order to update parameters in the training process accompanied by LSTM, Kalman filter is required to have adaptive filtering effect. So, we introduced a time-varying weighting factor to update matrix Q and matrix R in each batch to help the parameters converge stably, which was also helpful to deal with the time series data of pollutant concentration with different changing trends.

LSTM layer: LSTM layer is used to learn the input sequence $X$, and the hidden layer state of LSTM is recorded as $h_t$ at time $t$. The formula represents as

$$h_t = [h_{t,1}, h_{t,2}, \cdots h_{t,i}, \cdots h_{t,T}] \ i \in [1, T] \tag{31}$$

$$h_{t,i} = LSTM(x_{t,i}, h_{t-1,i}) \ i \in [1, T] \tag{32}$$

Attention layer: the input of the attention layer is the output $h_t$ of the previous layer, attention weight is $\alpha_{t,i}$ and the output of this layer is $S_t$. The formula represents as

$$S_t = \sum_{i=1}^{T} \alpha_{t,i} h_{t,i} \tag{33}$$

Output layer: The fully connected layer whose activation function is sigmoid was selected to output the predicted value Y of pollutant concentration at $t + 1$ moment. It is the result of second prediction of air quality and is represented as

$$y = sigmoid(\omega S_t + b) \tag{34}$$

In the training process of the model, the new Kalman gain $K_t$ and the new noise covariance matrix $P_t$ set off the backpropagation of LSTM according to the gradient descent direction of LSTM and to update the Kalman gain. The noise covariance matrix $P_t$ will update according to the new Kalman gain as formula (21) describes to help Kalman filter to prepare for the next batch of training. The update of the Kalman gain will be placed after the backpropagation (gradient-descent algorithm) and is presented as follows

$$\frac{dP_t}{dK_t} = \frac{d((I - K_t H)P_t^-(I - K_t H)^T + K_t r K_t^T)}{dK_t} = 2(I - K_t H)P_t^-\left(-H^T\right) + 2K_t R \tag{35}$$

According to the idea of optimization, we set $\frac{dP_t}{dK_t} = 0$; in this case, the error value of the optimal estimation is minimum, and the Kalman gain is updated according to Formula (20), which will be updated with each LSTM parameter update. As a submodule of Kalman-LSTM-attention model, the change in parameters update order of Kalman filter have no effect on the state optimal estimation and model prediction results.

This chapter introduces the principle and structure of the main model in this paper. Dynamic filtering of Kalman filter is introduced as a highly reliable data fusion, which effectively combines the pollutant concentration monitored by sensors with the first forecast data of WRF-CMAQ system. By adding the attention mechanism to the classical LSTM structure, the ability of the Kalman-attention-LSTM system to capture temporal information features is improved.

## 3. Results

### 3.1. Analysis of Correlation Coefficient and Cluster Characteristics of Pollution Data

The data of five meteorological conditions (temperature, humidity, air pressure, wind direction, and wind speed) and six pollutants ($SO_2$, $NO_2$, $PM_{10}$, $PM_{2.5}$, $O_3$ and CO) obtained by hourly measurement were preprocessed. The linear fitting of correlation coefficients of two variables in horizontal and vertical coordinates of each other is shown in Figure 8. Each small figure in Figure 8 is drawn using the distribution of sample points corresponding to two different variables (from five meteorological conditions or six pollutants) at the same time. The linear fitting results of the variation trends of these sample points are shown in the small figure with red lines. The respective data sets of both of the two variables were used to construct the covariance matrix, and 121 correlation coefficients were finally calculated as shown in Figure 9. The correlation coefficient in Figure 9 is essentially the slope of the linear fitting in Figure 8. The correlation coefficient can reflect the degree of independent influence between two variables. The symbol before the value of correlation coefficient $r$ represents the direction of influence between two variables, the plus sign represents positive correlation and the minus sign represents negative correlation. The absolute value of the correlation coefficient is between 0 and 1. Generally speaking, the closer $r$ is to 1, the stronger the correlation degree between the two variables will be. Conversely, the closer $r$ is to 0, the weaker the correlation degree between the two quantities will be.

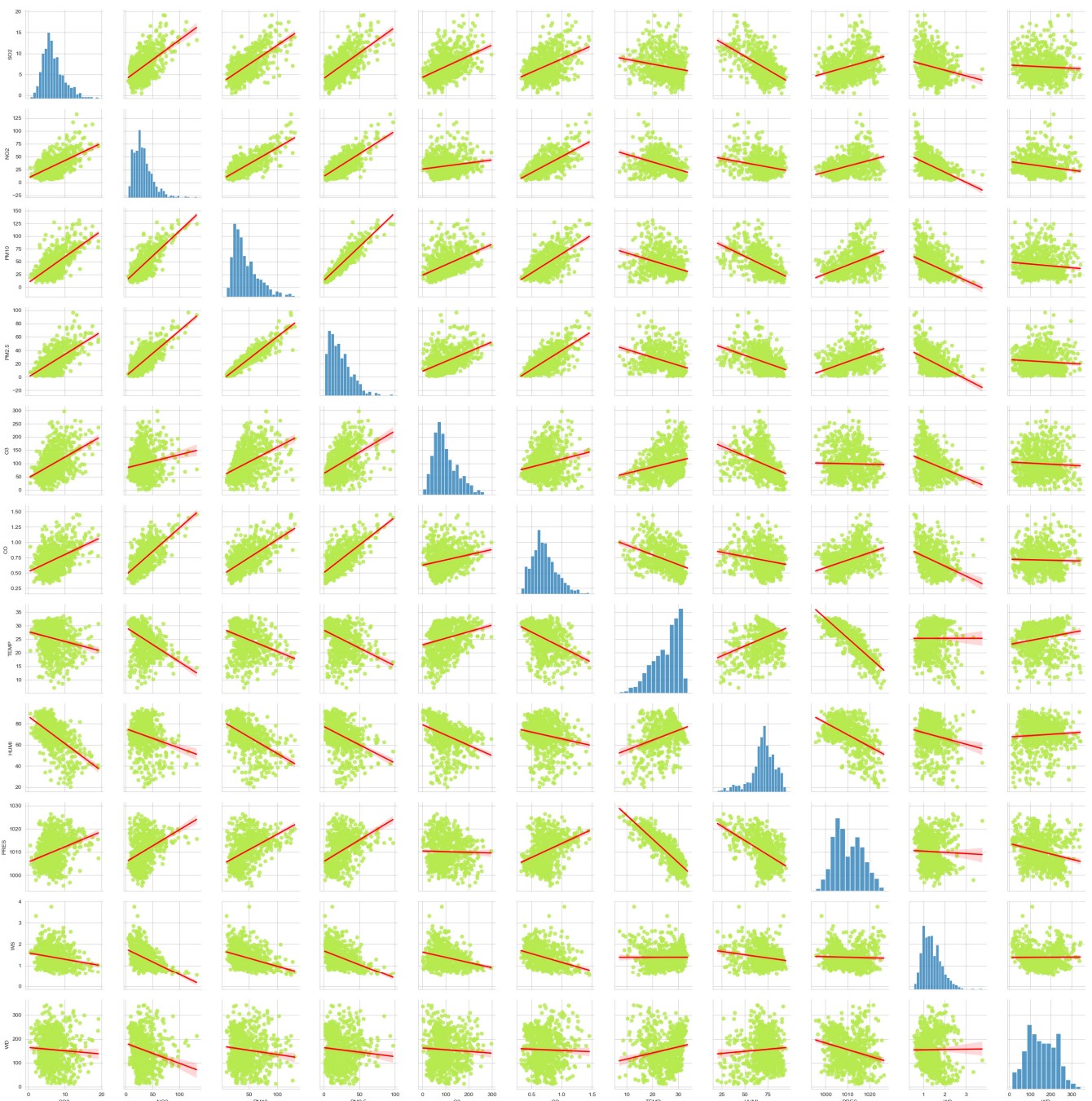

**Figure 8.** Linear fitting of correlation coefficients.

After clustering the data of the concentration of six pollutants (SO$_2$, NO$_2$, PM$_{10}$, PM$_{2.5}$, O$_3$ and CO), they are shown in Figure 10a.

The clustering of measured meteorological data (temperature, humidity, air pressure, wind direction and wind speed) are shown in Figure 10b.

The cluster centers of K-means are selected as two centers according to the two changed states in the AQI, showing either increase or decrease, which are distinguished by red dots and green triangle points in Figure 10.

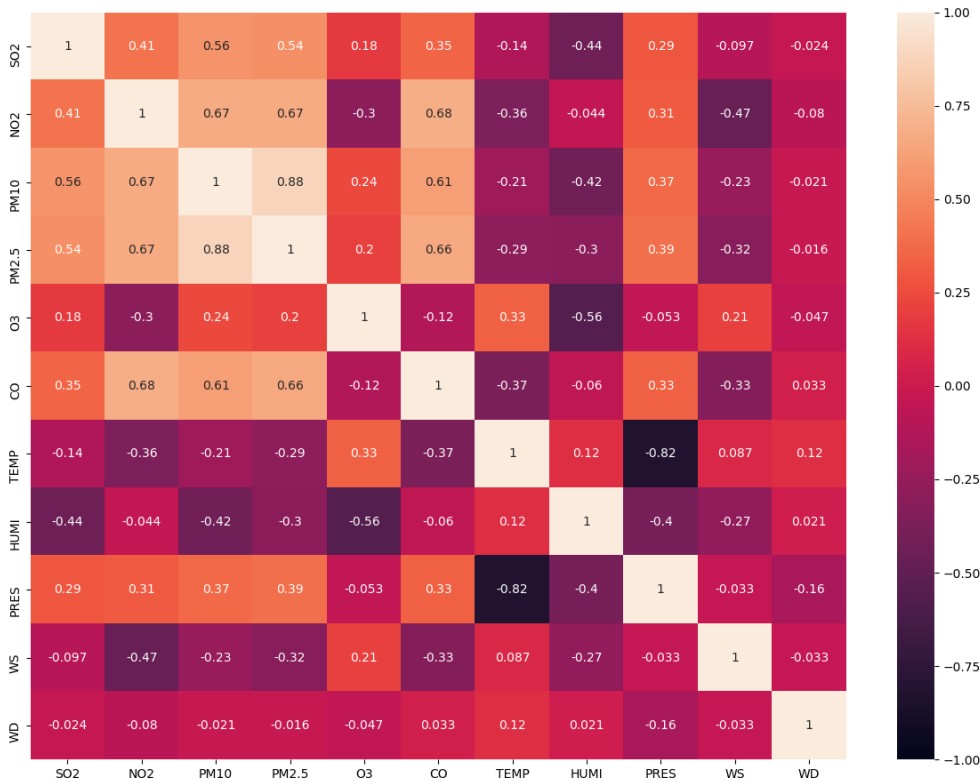

**Figure 9.** Correlation coefficient thermal diagram.

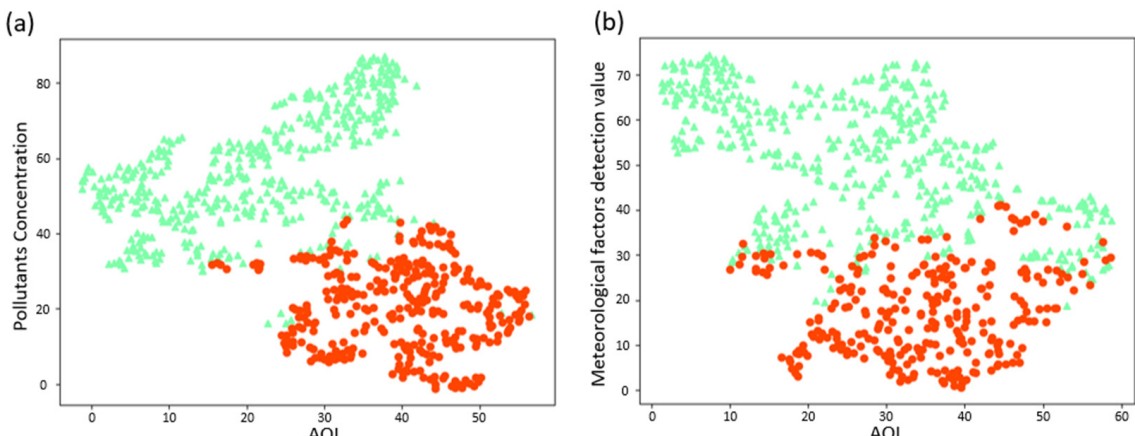

**Figure 10.** Clustering results: (**a**) clustering effect of concentration data of six pollutants; (**b**) clustering effect of measured meteorological data. Red dots represent the increase status of AQI, green triangle points the decrease status of AQI.

We use the contour coefficient to measure the clustering effect of the K-means clustering algorithm. The formula of the contour coefficient represents as

$$s = \frac{y - x}{\max(x, y)} \tag{36}$$

where $x$ is the distance of the vector from all the other points in the cluster to which it belongs, and represents the minimum average dissimilarity of the vector compared with the other clusters; $y$ is the average distance of a vector from all points in a cluster that do not contain it and represents the average degree of dissimilarity between a vector and other points in the same cluster; the $s$ range is limited to $(-1, 1)$; the plus sign means more similar

to samples in the cluster; and the minus sign means more similar to samples outside the cluster, with $|s|$ representing the degree of similarity.

The contour coefficient of the sample cluster for the concentration data of six major pollutants was 0.369. The contour coefficient of the clustering effect of measured meteorological data samples is 0.096.

The closer the contour coefficient is to 0, the lower the impact of the current clustering features on the AQI. The coefficient of 0.096 and 0.369 indicates that meteorological features (temperature, humidity, air pressure, wind direction, and wind speed) have a low impact on the AQI; however, six pollutant concentrations ($SO_2$, $NO_2$, $PM_{10}$, $PM_{2.5}$, $O_3$ and CO) have a high impact on the AQI. Therefore, the determinants of the AQI should be found out via the interaction of pollutant concentrations. Based on the calculation results of Equations (5)–(7), the IAQI value of $O_3$ is the maximum and is much larger than other pollutants for most days, meaning that the concentration of $O_3$ has the greatest determining effect on the value of the AQI. Moreover, in Figure 9, the correlation coefficient between $O_3$ and other pollutant information is the smallest, which means that $O_3$ will not be affected easily by other pollutants. Therefore, $O_3$ is selected as the most important variable concerning pollutant concentration information.

### 3.2. Kalman Filter Fitting Effect

The output result of the Kalman filter is the data fusion result of a pollutant concentration predicted for the first time and measured using pollutant concentration. The construction of a one-dimensional array, which contains data on the AQI and six pollution indicators ($SO_2$, $NO_2$, $PM_{10}$, $PM_{2.5}$, $O_3$ and CO), has seven characteristics.

The data fusion effect of the Kalman filter with six different pollutant concentrations: (a) $SO_2$, (b) $NO_2$, (c) $PM_{10}$, (d) $PM_{2.5}$, (e) $O_3$ and (f) CO are shown in Figure 11. The Kalman filter has a time granularity of one day. In Figure 11, the abscissa represents the daily sample points and the ordinate represents the pollutant concentration value. Furthermore, the blue curve represents the real measurement curve from the monitoring site, the black dots represent the first prediction curve from the monitoring site and the red curve represents the fitting result output of the Kalman filter on the two groups of input data. The output results proved that the Kalman filter with appropriate parameters can provide an ideal data fusion effort for Kalman-attention-LSTM network prediction.

### 3.3. Experimental Environment and Parameter Settings

After data pre-processing of the first prediction result generated using the WRF-CMAQ model and the actual monitoring data from monitoring site, the pre-process results (including the two input groups mentioned above) are made into a data set for second prediction; next, the data set was divided into a training set (70%), a validation set (10%) and a test set (20%). This data set will be used as the training and prediction material for the Kalman-attention-LSTM model. The training sets and validation sets are generated using random sampling, rather than partitioning, to ensure data consistency.

The input and output time granularity of the Kalman-attention-LSTM is one day, so the time granularity of the corrected sample data used for training and prediction is one day.

We adopted PyTorch on Windows as our experimental environment. Some other development tools, such as Python, NumPy and the d2l library were used in our experiments. The detailed hardware configurations and software versions are shown in Table 1.

The training parameter settings of the model are shown in Table 2. Input size represents the characteristic dimension of the input data. Hidden size represents the dimension of the hidden layer in LSTM. Num layer represents the number of layers in a recurrent neural network. Batch size represents the number of samples used in one iteration. Loss function use L2 loss. Learning rate represents the magnitude of each parameter update. Epoch ensures all training samples in the training set are trained and learned once. Each time step is run, the parameter weight is updated once, which means that learning is carried

out. Each parameter update requires batch size samples for operation learning, and the parameters are adjusted and updated once according to the operation results.

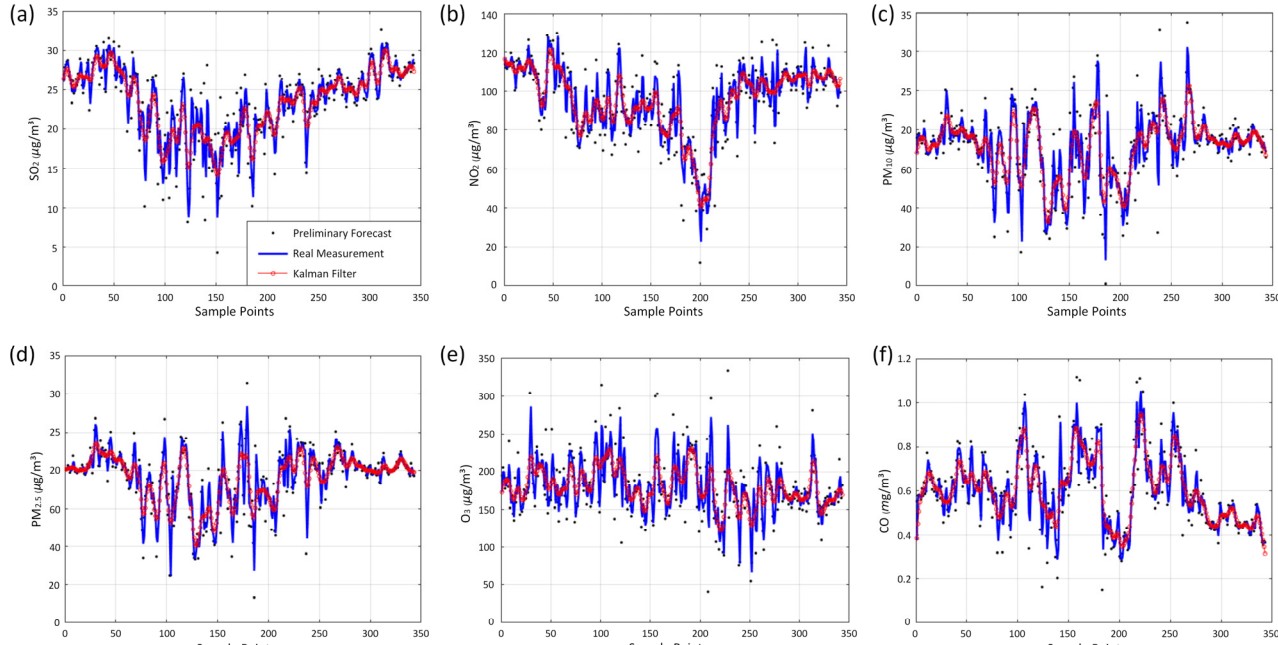

**Figure 11.** The fitting effect of the Kalman filter on the concentration of six pollutants: (**a**) SO$_2$; (**b**) NO$_2$; (**c**) PM$_{10}$; (**d**) PM$_{2.5}$; (**e**) O$_3$ and (**f**) CO.

**Table 1.** The detailed hardware configurations and software versions.

| Hardware and Software | Details |
|---|---|
| CPU | AMD Ryzen 9 5950X |
| GPU | Nvidia GeForceRTX3090 |
| RAM | 64 G |
| Hard disk | 1 TB |
| Operating system | Windows 10 Professional |
| Development tools | VSCode (Jupyter Notebook) |
| Development language | Python 3.8.12 |
| Deep learning framework | PyTorch |
| Other libraries | NumPy, d2l |

**Table 2.** The training parameter settings for the model.

| Hyperparameter | Details |
|---|---|
| Input size | 11 |
| Hidden size | 16 |
| Num layer | 1 |
| Batch size | 128 |
| Loss function | L2 loss |
| Learning rate | 0.01 |
| Epoch | 50 |
| Time step | 48 |
| Bias | True |
| Batch first | True |
| Dropout | 0.1 |
| Bidirectional | False |

During the experiment, we noticed that the LSTM model had a certain degree of gradient disappearance in the training process for some datasets. For this problem, usually the LSTM-forgetting gate value can be selected between 0 and 1 (sigmoid activation function). We chose to make this value close to 1 to saturate the forgetting gate. At this point, the long-distance information gradient does not disappear, and the gradient can be well transmitted in the LSTM, which greatly reduces the probability of gradient disappearing. In addition, we tried to use the Softsign activation function to replace Tanh, which is faster and helpful to overcome the vanishing gradient problem in the LSTM. Furthermore, we used the L2 regularization algorithm to prevent overfitting of the LSTM network. The L2 constraint usually imposes a large penalty on sparse weight vectors with spikes while preferring uniform parameters. This will encourage neural units to make use of all inputs from the upper layer, rather than just some of them. Therefore, after the addition of the L2 regularization algorithm, weight decay makes the network prefer to learn relatively small weights. $\theta$ is the parameter of the network layer to be learned, $\lambda$ controls the size of the regular term, and is presented as

$$L(\theta) = L(\theta) + \frac{\lambda}{2}||w||^2 \tag{37}$$

### 3.4. Second Prediction Results

The comparison between the second prediction results, first prediction values and the real measure values for $O_3$ data is shown in Figure 12.

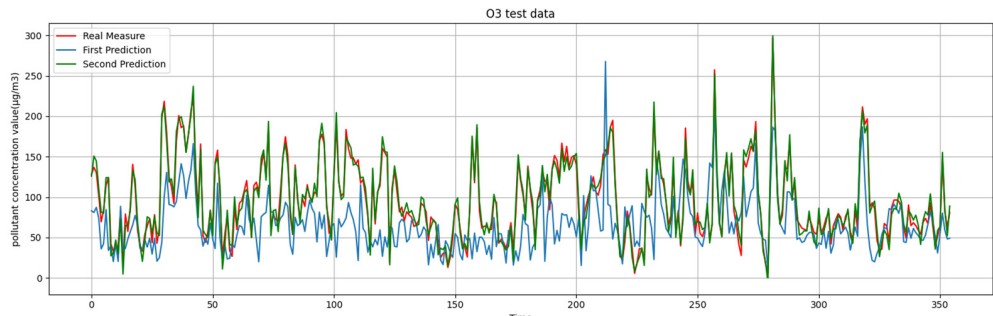

**Figure 12.** The second prediction results, first prediction values and the real measure values for $O_3$.

The second prediction results for $SO_2$ data are shown in Figure 13. The second prediction results for $PM_{10}$ data are shown in Figure 14. The second prediction results for $PM_{2.5}$ data are shown in Figure 15. The second prediction results for $NO_2$ data are shown in Figure 16. The second prediction results for CO data are shown in Figure 17.

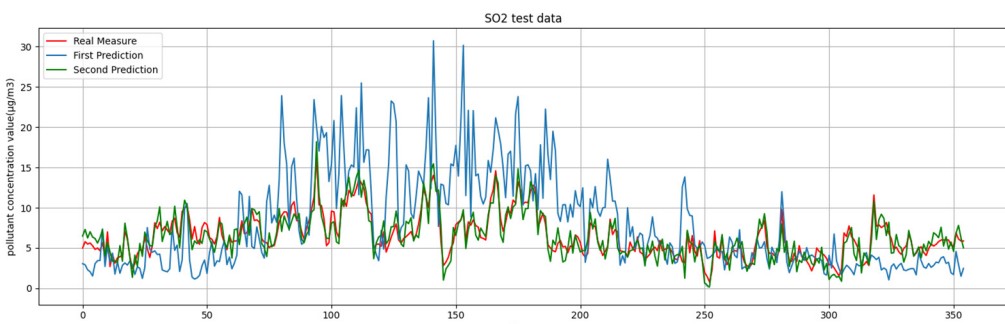

**Figure 13.** The second prediction results, first prediction values and the real measure values for $SO_2$.

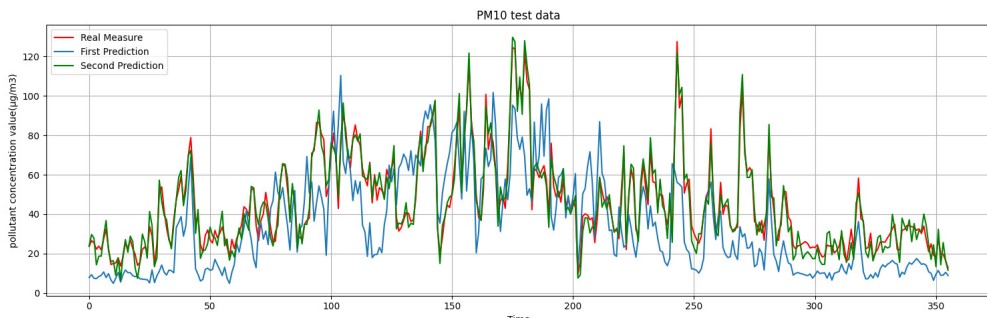

**Figure 14.** The second prediction results, first prediction values and the real measure values for PM$_{10}$.

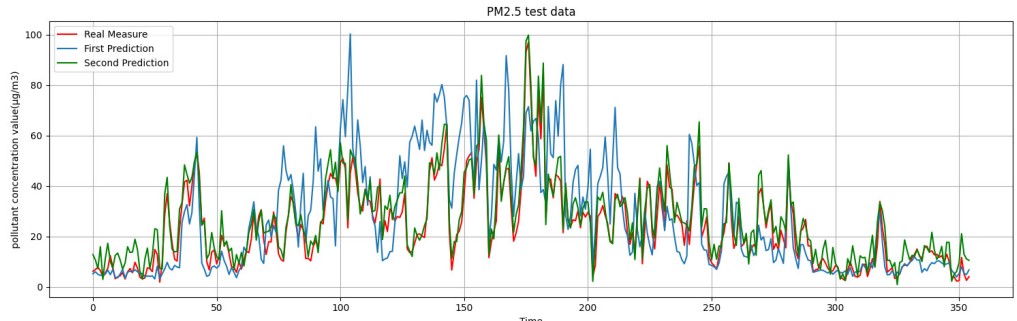

**Figure 15.** The second prediction results, first prediction values and the real measure values for PM$_{2.5}$.

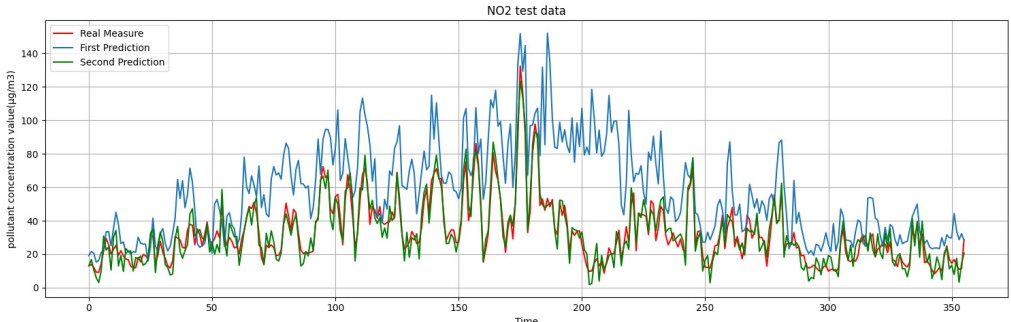

**Figure 16.** The second prediction results, first prediction values and the real measure values for NO$_2$.

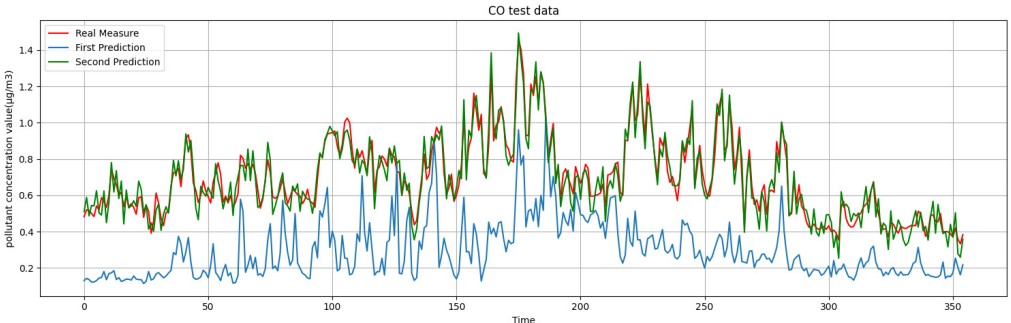

**Figure 17.** The second prediction results, first prediction values and the real measure values for CO.

In Figures 12–17, the abscissa represents the passage of time every day and the ordinate represents the concentration value of pollutants in this figure. Each figure contains three curves: real measure values, first prediction values and second prediction results. According to the analysis and conclusion above, O$_3$ has a major impact on the AQI, and the prediction curve of O$_3$ is equivalent to the prediction curve of the AQI.

### 3.5. Model Performance Evaluation and Algorithm Comparison

Performance analysis and evaluation of linear regression algorithm model usually rely on the standard error (SE), root mean squared error (RMSE), mean absolute error (MAE) and the R-square.

The standard error (SE) is used to predict the accuracy of the sample data. The smaller the standard error is, the smaller the gap between the sample mean and the population mean is and the more representative the sample data is of the population. $\hat{y}_i$ represents the predicted value, $y_i$ represents the real measure value and n represents the number of samples, and its calculation method is shown in Equation (38):

$$SE = \sqrt{\frac{\sum_{i=1}^{n}(\hat{y}_i - y_i)^2}{n(n-1)}} \tag{38}$$

The RMSE is the square root of the ratio of the square of the deviation between the real measure value and the predicted value, and its calculation method is shown in Equation (39). The RMSE is more sensitive to outliers in the data. The use of the RMSE as an evaluation index magnifies the gap between large errors, and the smaller the value of the RMSE in the measurement, the greater the model's ability to fit data is. $\hat{y}_i$ represents the predicted value, $y_i$ represents the real measure value and n represents the number of samples.

$$RMSE = \sqrt{\frac{\sum_{i=1}^{n}(y_i - \hat{y}_i)^2}{n}} \tag{39}$$

The RMSE has the same dimension as the MAE, but the RMSE is larger than the MAE. The MAE reflects the true error. The MAE calculation method is shown in Equation (40). $\hat{y}_i$ represents the predicted value, $y_i$ represents the real measure value and n represents the number of samples.

$$MAE = \frac{1}{n}\sum_{i=1}^{n}|y_i - \hat{y}_i| \tag{40}$$

The best indicator to measure the linear regression method is R-square, which represents the size of the model fitting ability. The R-square calculation method is shown in Equation (41). The larger the value, the better fitting effect. $\hat{y}_i$ represents the predicted value, $y_i$ represents the real measure value, $\overline{y}_i$ represents the average value of $y_i$ and $n$ represents the number of samples.

$$R^2 = 1 - \frac{\sum_{i=1}^{n}(y_i - \hat{y}_i)^2}{\sum_{i=1}^{n}(y_i - \overline{y}_i)^2} \tag{41}$$

In order to compare the effects of the Kalman-attention-LSTM model and other traditional time series prediction models, such as the RNN, GRU, and LSTM, we use the original $O_3$ pollutant concentration data set to train the above four prediction algorithms respectively, and calculate the values of the RMSE, MAE, and R-square. At the same time, we consider that $O_3$ has a decisive influence on the AQI, and that the pollutant concentration value of $O_3$ has the most predictive value.

In Table 3, the Kalman-attention-LSTM model improved significantly, compared with the six models. In the first prediction (from the WRF-CMAQ), for the RNN, GRU, LSTM, attention-LSTM and Kalman-LSTM, the SE improved by 83.26%, 51.64%, 43.58%, 45% and 29%, respectively; the RMSE improved by 83.16%, 51.52%, 43.21%, 44.59%, 26.07% and 28.32%, respectively; the MAE improved by 80.49%, 56.96%, 46.75%, 49.97%, 26.04%and 27.36%, respectively; and the R-square improved by 85.3%, 16.4%, 10.3%, 11.5%, 2.7% and 3.3%, respectively. As shown in Table 3, the SE, RMES, MAE and R-square indicate that the results of the first prediction (from the WRF-CMAQ) do not reflect the value of pollutant concentration, but only reflect the general trend of pollutant concentration. The reason for the inaccurate prediction is also due to the unique mechanism of the WRF-CMAQ model,

which is subject to the uncertainty of the simulated meteorological field and emission inventory, as well as the incomplete clarity of the generation mechanism of pollutants. The results of the WRF-CMAQ prediction model are not ideal. Therefore, a second prediction has special significance for improving the accuracy of weather forecast.

**Table 3.** $O_3$ prediction error comparison of different models.

| Network Model | SE | RMSE | MAE | R-Square |
|---|---|---|---|---|
| First prediction (from WRF-CMAQ) | 2.63 | 50.19 | 37.42 | 0.115 |
| RNN | 0.91 | 17.43 | 16.96 | 0.804 |
| GRU | 0.78 | 14.88 | 13.71 | 0.865 |
| LSTM | 0.80 | 15.25 | 14.59 | 0.853 |
| Attention-LSTM | 0.60 | 11.43 | 9.87 | 0.941 |
| Kalman-LSTM | 0.62 | 11.79 | 10.05 | 0.935 |
| Kalman-attention-LSTM | 0.44 | 8.45 | 7.30 | 0.968 |

In addition, we used different models to predict six different major pollutants, and the prediction results are respectively shown in Tables 3–8. By comparing the SE, RMSE, MAE and R-square, we can draw the following conclusions: The second prediction method proposed in this paper (by using the Kalman-attention-LSTM model) has significantly improved the prediction accuracy compared with the classical time series prediction and primary prediction results. The WRF-CMAQ model, which provides the first prediction result, is far from meeting the prediction demand. There is no doubt that the second prediction is necessary for the prediction of pollutant concentration, and the combined effect of the Kalman filter and attention mechanism improves the accuracy of this model.

**Table 4.** $SO_2$ prediction error comparison of different models.

| Network Model | SE | RMSE | MAE | R-Square |
|---|---|---|---|---|
| First prediction (from WRF-CMAQ) | 3.27 | 62.38 | 45.41 | 0.063 |
| RNN | 2.16 | 19.65 | 17.83 | 0.785 |
| GRU | 0.86 | 16.37 | 14.71 | 0.843 |
| LSTM | 0.79 | 15.12 | 13.88 | 0.856 |
| Attention-LSTM | 0.67 | 12.74 | 11.39 | 0.923 |
| Kalman-LSTM | 0.68 | 12.95 | 11.64 | 0.909 |
| Kalman-attention-LSTM | 0.58 | 11.14 | 9.74 | 0.944 |

**Table 5.** $PM_{10}$ prediction error comparison of different models.

| Network Model | SE | RMSE | MAE | R-Square |
|---|---|---|---|---|
| First prediction (from WRF-CMAQ) | 2.92 | 55.89 | 41.57 | 0.092 |
| RNN | 0.91 | 17.54 | 15.65 | 0.828 |
| GRU | 0.78 | 15.02 | 13.32 | 0.863 |
| LSTM | 0.76 | 14.76 | 13.12 | 0.874 |
| Attention-LSTM | 0.62 | 11.78 | 10.11 | 0.931 |
| Kalman-LSTM | 0.64 | 12.25 | 10.92 | 0.922 |
| Kalman-attention-LSTM | 0.53 | 10.09 | 8.77 | 0.951 |

**Table 6.** $PM_{2.5}$ prediction error comparison of different models.

| Network Model | SE | RMSE | MAE | R-Square |
|---|---|---|---|---|
| First prediction (from WRF-CMAQ) | 3.36 | 64.22 | 48.31 | 0.056 |
| RNN | 0.88 | 16.83 | 15.97 | 0.832 |
| GRU | 0.80 | 15.34 | 13.93 | 0.851 |
| LSTM | 0.79 | 15.21 | 14.04 | 0.854 |
| Attention-LSTM | 0.68 | 12.99 | 11.51 | 0.916 |
| Kalman-LSTM | 0.70 | 13.40 | 12.11 | 0.903 |
| Kalman-attention-LSTM | 0.61 | 11.69 | 9.98 | 0.935 |

**Table 7.** NO$_2$ prediction error comparison of different models.

| Network Model | SE | RMSE | MAE | R-Square |
|---|---|---|---|---|
| First prediction (from WRF-CMAQ) | 3.13 | 59.82 | 47.11 | 0.077 |
| RNN | 0.9 | 17.21 | 16.84 | 0.814 |
| GRU | 0.77 | 14.66 | 12.90 | 0.875 |
| LSTM | 0.78 | 14.93 | 13.50 | 0.868 |
| Attention-LSTM | 0.62 | 11.76 | 10.11 | 0.933 |
| Kalman-LSTM | 0.58 | 11.15 | 9.77 | 0.941 |
| Kalman-attention-LSTM | 0.47 | 8.98 | 8.03 | 0.962 |

**Table 8.** CO prediction error comparison of different models.

| Network Model | SE | RMSE | MAE | R-Square |
|---|---|---|---|---|
| First prediction (from WRF-CMAQ) | 3.04 | 58.03 | 39.98 | 0.083 |
| RNN | 1.08 | 20.61 | 18.26 | 0.771 |
| GRU | 0.86 | 16.51 | 16.09 | 0.833 |
| LSTM | 0.89 | 16.94 | 15.88 | 0.838 |
| Attention-LSTM | 0.68 | 13.06 | 12.11 | 0.906 |
| Kalman-LSTM | 0.73 | 14.11 | 12.96 | 0.884 |
| Kalman-attention-LSTM | 0.65 | 12.33 | 11.01 | 0.927 |

## 4. Conclusions

In this paper, we proposed the innovative Kalman-attention-LSTM model, aiming to further improve the prediction accuracy of pollutant concentration and AQI on the basis of the traditional time series prediction model. The specific realization process of the model was as follows:

(1) First of all, data pre-processing is required. We filled in random gaps in weather data and normalized weather data of different orders of magnitude.

(2) Secondly, a cluster analysis was conducted on normalized pollutant concentration data and meteorological data. We determined the correlation coefficient between different pollutants and meteorological information, and identified the pollutant concentration information with the greatest influence on the AQI as O$_3$.

(3) Then, we chose the appropriate parameters for the Kalman filter to fuse the measured and first-prediction meteorological data, which intended to make the prediction more accurate and reliable in dynamic.

(4) Finally, the attention mechanism is used to set the weights of the inputs of different time segments in the traditional LTSM model and was intended to improve the prediction accuracy.

(5) In comparison with the traditional RNN, GRU, LSTM, attention-LSTM and Kalman-LSTM, the Kalman-attention-LSTM model shows better effort from the perspective of the SE, RMSE, MAE and R-square evaluations, which indicates that our Kalman-attention-LSTM model has a higher prediction accuracy in a single pollutant concentration prediction.

Compared with other models, our Kalman-attention-LSTM model has better indicators. In order to further evaluate the generalization ability of this model, we will use this model to predict and analyze cities under different spatio-temporal backgrounds in future. In addition, other pollutants in the air that cannot be ignored are aerosol pollutants, biological sources of aerosols in the air with a spatio-temporal distribution that is relatively complex and to a large extent influenced by other contaminants in the air. The existence of the air pollution index in a numerical prediction method will play an important role for subsequent research. We will also make predictions for aerosol pollutants, discussing the value of the model in the broadest possible context.

In addition, we still believe that hybrid models such as the Kalman-attention-LSTM model play a positive role in improving the prediction accuracy. In future works, more

focus will be laid on how to expand the range of application for the integrated model and to improve the accuracy of various data prediction due to the integration of the advantages of hybrid models, for example, integrating CNN into our model to extract more reliable spatial distribution for forecasting, or expanding the range of application for the integrated LSTM and Kalman filter in order to improve the accuracy of various data prediction.

**Author Contributions:** Methodology, H.Z. (Hao Zhou), T.W. and H.Z. (Hongchao Zhao); software, H.Z. (Hongchao Zhao) and Z.W.; supervision, H.Z. (Hao Zhou); validation, H.Z. (Hao Zhou) and T.W.; visualization, H.Z. (Hao Zhou) and Z.W.; writing—original draft, H.Z. (Hao Zhou) and T.W.; writing—review and editing, H.Z. (Hao Zhou), T.W. and H.Z. (Hao Zhou). All authors have read and agreed to the published version of the manuscript.

**Funding:** This research received no external funding.

**Institutional Review Board Statement:** Not applicable.

**Informed Consent Statement:** Not applicable.

**Data Availability Statement:** Not applicable.

**Acknowledgments:** The authors thank Yuqiao Xu and Wanting Li for their contributions to the preliminary research of this article. The authors thank Gang Yuan and Yunhao Peng for providing us with academic guidance and detailed grammar revision work. Thanks to the experts for their evaluation and suggestions on our method.

**Conflicts of Interest:** The authors declare no conflict of interest.

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
