# Peer review of "Updated Prediction of Air Quality Based on Kalman-Attention-LSTM Network"

_sustainability, doi:10.3390/su15010356_

Round 1
Reviewer 1 Report (Previous Reviewer 1)
Der Authors,
The revision you made is considered as satisfactory. I have no further comments on your work.
I wish you the best of luck
Best regards
Author Response
Thank you very much for your comments. It means a lot to us. Good luck to you too.
Reviewer 2 Report (Previous Reviewer 3)
This manuscript modifies the description of data division. However, the description is clear in previous versions. Further, I have not noticed the strategy to embed Kalman filter into deep learning framework that I suggested the authors to modify in previous review. If the strategy can not be designed, the proposed model lacks originality. In this manuscript, Kalman filter is still a preprocessing step for the prediction model. Therefore, this manuscript has not been improved significantly.
Author Response
Thanks for your valuable comments. We redesigned the entire structure of the network (P13, Figure 7. Architectures of the Kalman-Attention-LSTM network). In addition, considering that in the previous design, the parameters of Kalman algorithm are fixed (mainly the matrix Q and matrix R are selected artificially), and Kalman filter only participates in the forward propagation of the overall model (including LSTM) training process, it did not participate in back propagation to update the Kalman filter parameters, which is misleading. Therefore, we adjusted the structure as follows:
1.During the training process of the model, the update of Kalman gain and noise covariance matrix was adjusted to the back propagation of LSTM. Kalman gain was updated according to the gradient descent direction of LSTM, and then the noise covariance matrix was further updated to prepare for the call of Kalman filtering in the training of the next batch. (P13, 2.2.6. Kalman-Attention-LSTM Network)
2.We introduced a weighting factor that changes with time and updated matrix Q and matrix R in each iteration (Batch) of LSTM parameters, so that when the parameters tend to converge stably, an adaptive filtering effect is formed, which is also helpful to deal with the time series data of pollutant concentration with different changing trends. (P12, 2.2.6. Kalman-Attention-LSTM Network)
We believe that it is reasonable to embed Kalman Filter into LSTM in the above way, which means that Kalman Filter is a part of data forward propagation in the Kalman-Attention-LSTM model, while back propagation updates the parameters of LSTM and Kalman Filter at the same time.
Reviewer 3 Report (New Reviewer)
Reviewer Report for Secondary Prediction of Air Quality based on Kalman-Attention-LSTM Network submitted to Sustainability.
The author of the manuscript proposed a deep learning algorithm based on LSTM, attention, and Kalman Filter to forecast air pollution. The empirical evidence shows that the proposed model has a high forecasting power compared to the other well-known deep learning algorithms. The general view of the paper is good and it would be a part of the air pollution forecasting literature. Please consider the following
Please increase the resolution of the figures.
In the literature please consider https://doi.org/10.1007/s13762-021-03730-3 which used hybrid algorithms to forecast air pollutants.
I couldn’t see the architecture of the proposed methodology for different cases. If possible could you please provide the hyperparameters and the cases for the proposed methodology?
Could you please mention how you prevent overfitting?
In the introduction please emphasize the need for the Kalman filter algorithm in depth.
Generally, the manuscript is written in a very systematic way and the quality of the manuscript is very good.
Author Response
(1) Thanks for your reminder, we have updated the figures with high-definition quality and uploaded FIGURES.zip
(2) Thanks for providing this paper. We referred to it and included it to show the gap between these studies and the proposed one in our paper (P2, 1.2. Related Works, [4]).
(3) We updated Figure 7 (P13, Figure 7. Architectures of the Kalman-Attention-LSTM network) to present a clearer Architectures of the Kalman-Attention-LSTM network. We used the same network structure for different cases. To be specific, during the time series data of six different pollutant (O3, SO2, PM10, PM2.5, NO2, CO) concentrations prediction, they are all includes the same climate data(temperature, humidity, air pressure, wind direction, wind speed), so the data set is trained and predicted by the Kalman-Attention-LSTM model with the same structure in Figure 7, and we use the same model hyperparameters for the six pollutant concentrations prediction because the distribution feature of the six data is relatively balanced, and there is a partial strong correlation between the changing trends of different pollutant concentrations. In some way, the types of variables and the dimensions of the variables are similar. Hyperparameters are given in Table 2 (P16, Table 2. The training parameter settings of the model).
Besides, we use the Pytorch framework to build the LSTM network. The contents of Table 2 are selected based on Pytorch's definition of LSTM (https://pytorch.org/docs/stable/generated/torch.nn.LSTM.html#torch.nn.LSTM):
Parameters:
input_size – The number of expected features in the input x
hidden_size – The number of features in the hidden state h
num_layers – Number of recurrent layers. E.g., setting num_layers=2 would mean stacking two LSTMs together to form a stacked LSTM, with the second LSTM taking in outputs of the first LSTM and computing the final results. Default: 1
bias – If False, then the layer does not use bias weights b_ih and b_hh. Default: True
batch_first – If True, then the input and output tensors are provided as (batch, seq, feature) instead of (seq, batch, feature). Note that this does not apply to hidden or cell states. See the Inputs/Outputs sections below for details. Default: False
dropout – If non-zero, introduces a Dropout layer on the outputs of each LSTM layer except the last layer, with dropout probability equal to dropout. Default: 0
bidirectional – If True, becomes a bidirectional LSTM. Default: False
proj_size – If > 0, will use LSTM with projections of corresponding size. Default: 0
(4) Thanks to the following methods to avoid the overfitting: Firstly, we used L2 regularization algorithm to prevent overfitting of the LSTM network (P16, 3.3. Experimental Environment And Parameter Settings). Secondly, the training sets and validation sets are generated by random sampling, rather than partitioning, to ensure data consistency (P16, 3.3. Experimental Environment And Parameter Settings). Besides, we set dropout rate = 0.1 (P16, Table 2. The training parameter settings of the model). Last but not least, the data normalization is completed in the pre-processing step. (P6, 2.1. Data Collection And Preprocessing).
(5) We made further discuss about the role and necessity of the Kalman filter (P2, 1.1. background information).
This manuscript is a resubmission of an earlier submission. The following is a list of the peer review reports and author responses from that submission.
Round 1
Reviewer 1 Report
The article entitled “Secondary Prediction of Air Quality based on Kalman-Attention-LSTM Network”, presents a secondary prediction method of air pollutant concentration based on Attention-LSTM (Attention Long Short-Term Memory) model and Kalman Filter.
The subject of this research work is utterly interesting, and although the subject of predicting air quality based on LSTM-Kalman Model is not entirely novel the authors manage to justify the contribution of their work by referring to an adequate number of related works and pointing to the new points of the approach they propose in this paper.
The methodology which is followed in this work is clearly defined and assessed adequately permitting other researchers to reproduce certain aspects. Additionally, the methodology analysis, as well as the assessment results are enriched with an efficient number of properly presented figures, tables and charts.
The conclusions of the research and their association with the results are satisfactory defined. Nevertheless, the authors are advised to include a discussion section wherein the results will be thoroughly interpreted in perspective of the working hypotheses, and the findings of the research as well as their implications will be discussed in the broadest context possible. Moreover, some future directions would be interesting to be included in the conclusions section.
Finally, the paper is well-structured in general and written in appropriate English language according to the standards of the Journal, however some minor spell-checking is required and the in-text referencing should be adjusted to the standards of the journal.
Author Response
Thank you for your comments.
(1) We add in-depth discussion on the research results as well as prospects for future research directions in the Conclusion chapter. (P20, 4. Conclusions)
(2) We carefully revised the paper and substantially improved the English writing while keeping the content unchanged. We have also double checked the language to avoid typos and mistakes. Besides, we have adjusted the in-text referencing to the standards of the journal. Now we believe that this version meets the requirements for publication in Sustainability.
Reviewer 2 Report
The study is very interesting.
The quality of the figures needs to be improved.
Why the authors use only three indices in evaluating the model's performances. The authors suggested adding Standard Error, if possible.
The authors can refer to a review paper published in the same area: DOI 10.1007/s11270-021-04989-5
There are some studies that have not included, maybe the authors can include them and show what is the gap between these studies and the proposed one in this paper: 10.1080/19942060.2021.1926328
Author Response
(1) Thanks to Reviewer for reminder, we have updated the figures with high-definition quality.
(2) We think this is an excellent suggestion. We reviewed relevant literature and found that common model performance evaluation indicators of linear regression problems include MSE, RMSE, MAE, R-Square, etc. In addition, we think that Standard Error can reflect the accuracy of predicted data, and we added this part in Table3-Table8.
(3) Thank the reviewers for providing two papers. We referred to them and included them to show the gap between these studies and the proposed one in our paper. (P2, 1.2. Related Works)
Reviewer 3 Report
The structure of this paper is clear. In the manuscript, Kalman filter is used to fuse the data and Attention-LSTM Network model is used for AQI prediction.
However, the filter and the model are two independent data processing steps, that lacks the orginality in AQI prediction model. The authors should design a strategy incorporating Kalman filter into Attention-LSTM Network model as a submodule. Kalman Filter is used to preprocess the data and is not closely related with Attention-LSTM model.
Author Response
We sincerely appreciate the valuable comments. And we want to address our study in the following points.
(1) In the concentration prediction of six pollutants, the prediction effect of Kalman-Attention-LSTM model is better than that of Attention-LSTM model, which is reflected in RMSE, MAE, R-Square and other evaluation parameters. It shows that as a submodule of this model, Kalman filter improves the prediction accuracy of pollutant concentration, and the role of Kalman filter is worthy of affirmation.
(2) The function of Kalman filter is different from the data smoothing effect of other commonly used filters, and it does not belong to the data preprocessing step. In fact, the function of the Kalman filter is to fuse the data of the meteorological information and the first predicted value, to make an optimal estimate of the true pollutant concentration state. As shown in Figure. 7, as a submodule of the overall model, the Kalman filter has self-iteration and prediction functions. (P12, Figure 7. Hierarchy of the Kalman-Attention⁃LSTM network.)
(3) Although both Attention-LSTM model and Kalman-LSTM model have been proposed by scholars, we have not found the application of another existing Kalman-Attention-LSTM model at present.
(4) The innovation of this paper is not only the proposal of Kalman-Attention-LSTM model, but also the further second prediction based on the first prediction model (WRF-CMAQ model). At present, few articles are related to the second prediction of pollutant concentration. And our results confirm that quadratic forecasting can greatly improve the prediction accuracy.
Reviewer 4 Report
This paper proposed a novel predictive model based on Attention-LSTM and Kalman filter. The novel method has been tested in air pollution prediction against a number of existing methods. I found this research interesting and the contribution of clear value. There are still some points need to be clarified in the manuscript. I would suggest revisions as listed herebelow:
1. How frequent is your observation in the KF? Do you have observations at every time steps
2. What is your initial error covariance in KF? And how this is chosen?
3. Fig 12-16 needs to be improved, the fontsize is too small for labels.
4. As an ablation test, it would be interesting to show the results of Kalman-LSTM (without attention)
5. An interesting perspective: how your model can be extended to high-dimensional dynamical systems for air pollution?
6. The literature review is good but the authors have missed out a number of contributions that already applied LSTM with data assimilation algorithms (including Kalman filter). Some references to build the related work:
Cheng, S., Prentice, I.C., Huang, Y., Jin, Y., Guo, Y.K. and Arcucci, R., 2022. Data-driven surrogate model with latent data assimilation: Application to wildfire forecasting. Journal of Computational Physics, p.111302.
https://pubs.rsc.org/en/content/articlehtml/2022/lc/d2lc00303a
https://arxiv.org/abs/2204.03497
However, to the best of the reviewer’s knowledge none of them have applied the attention mechanism. The authors may highlight this novelty.
Author Response
We sincerely appreciate for your valuable comments.
(1) The Kalman Filter has a time granularity of one day. (P15, 3.2. Kalman Filter Fitting Effect) After the previous data preprocessing steps, including Lagrange interpolation and data normalization, the daily data used by the Kalman filter is continuous, and the sbservation duration is one year. (P4, 2.1. Data Collection And Preprocessing)
(2) Usually, the initial error covariance in Kalman Filter represent as P0. The covariance represents the uncertainty of the state. To initialize the covariance matrix is to show some prior understanding of the state. The accurately determined covariance of the state is set to zero, and if there is no prior information we have to set a higher value. As the filter is updated, it decreases according to the measurement. So the P0 setting is not specified, it's an empirical parameter. According to our experiment, we generally set P0=k*I, k is a larger number, I is the identity matrix, eventually the matrix P will converge. The key factors affecting the performance of the filter are matrix Q and matrix R.
(3) We adjusted the format and size of Figure 12-16 to ensure that the details and labels in these figures are clear.
(4) We added the test results of Kalman-LSTM model (without Attention) in Table 3-8.
(5) High-dimensional dynamical systems often appear with nonlinear problems. Dynamical models involving systems of numerous differential equations are commonly used to describe meterological behavior. However, the Kalman filter and LSTM model used in this paper are not suitable for solving nonlinear complex multivariate dynamic models. In addition, this paper mainly focuses on linear regression, and the model evaluation methods we use include SE, RMSE, MAE, and R-Square. Again, these error analysis methods are only suitable for evaluating linear regression models. We focused on the data characteristics of local meteorological data and did not analyze the diffusion and dynamics models involving the surrounding area. In addition, we must admit that the authors' team's current understanding of this area is insufficient to support the model's expansion into this area.
(6) Thank the reviewers for providing three papers. We referred to them and included them to show the gap between these studies and the proposed one in our paper. (P4, 1.2. Related Works)
(7) In fact, the first combined use of the Attention Mechanism and the sequential model has been published as a conference paper at ICLR 2015, NEURAL MACHINE TRANSLATION BY JOINTLY LEARNING TO ALIGN AND TRANSLATE (https://arxiv.org/pdf/1409.0473.pdf). At present, Attention Mechanism has been widely used in the prediction model of deep learning methods. The essence of Attention Mechanism is as follows: For a given target, a weighted sum of the input is generated by a weight coefficient to identify which features in the input are important and which are not important to the target. In the concentration prediction of six pollutants, the prediction effect of Attention-LSTM model is better than LSTM model, which is reflected in RMSE, MAE, R-Square and other evaluation parameters.
Besides, The innovation of this paper is not only the proposal of Kalman-Attention-LSTM model or Attention Mechanism, but also the further second prediction based on the first prediction model (WRF-CMAQ model). At present, few articles are related to the second prediction of pollutant concentration. And our results confirm that quadratic forecasting can greatly improve the prediction accuracy. Conclusions about Attention Mechanism already exist. (Table 3-8 and P21, 4. Conclusions )
Round 2
Reviewer 3 Report
Just as the authors describe in this paper "The output data of the Kalman Filter are taken as the training and prediction sample of Attention-LSTM", Kalman Filter is a preprocessing step for the prediction and Attention-LSTM takes the results processed by Kalman Filter as input.
Author Response
(1) Thank you for pointing this out.
(2) It is true that we used Kalman Filter's output as the input of the next part in our model. But that's just an easy way to describe it. This is because both the original meteorological data and the first forecast data (From WRF-CMAQ model) have credibility problems. Intuitively speaking, Kalman filter plays a role of data fusion here. The current state can be estimated only by inputting the current measured value (data of 1 or more sensors) and the estimated value of the last cycle. The estimated current state comprehensively considers the sensor data (namely the so-called observed value and measured value) and the data of the previous state, which is the current optimal estimate. This estimated value can be considered the most reliable value. State estimation is to seek the best fitting state vector with observed data by mathematical method.
(3) We chose to present the effect of Kalman filter in the manuscript because we need to select a set of appropriate filter parameters (mainly the matrix Q and matrix R), debug and optimize the filter state estimation or the so-called filtering effect. Figure 11 shows the best filter effect in our experiment. However, in the actual experiment, we did not really use Kalman filter to generate a set of data to save, make a data set, and then give it to the LSTM network for training. The true experiment process is wait until the LSTM learning parameters are fixed, LSTM prediction and Kalman Filter state estimation are carried out synchronously. As submodules, they have direct data transmission and data call behaviors.
(4) Our pollutant prediction system is a linear, discrete and time-varying system with accurate and calculable information at each time step. Both the measured value and the first predicted value of each time step contain white noise. The above two points indicate that the prerequisite for using Kalman filter is perfectly met.
(5) We added the experimental results of Kalman-LSTM (without Attention) in Table 3-8. By comparing the results of LSTM and Kalman-LSTM, it can still be proved that Kalman Falter's advanced optimal state estimation is beneficial to improve the prediction accuracy.
(6) In conclusion, the advantages of Kalman filtering are as follows: the recursive method is adopted to solve the linear filtering problem. Only the current measured value and the estimated value of the previous sampling period can be used for state estimation. It does not require a large amount of storage space, and the calculation amount of each step is small and the calculation steps are clear, which is very suitable for computer processing.
Reviewer 4 Report
I would recommand publication for the revised manuscript
Round 3
Reviewer 3 Report
Thanks for the authors' responses. However, just as the authors write in the paper "The fitting result of pollutant concentration data output by Kalman Filter is made into a data set for secondary prediction and next, the data set was divided into a training set (70%), a validation set (10%), and a test set (20%).", it is obvious that Kalman Filter is a preprocessing step and the processed results are sent to Attention-LSTM model. Morever, this statement in the paper contradicts with the authors' response that "in the actual experiment, we did not really use Kalman filter to generate a set of data to save, make a data set, and then give it to the LSTM network for training.".